# Effective and Efficient Federated Tree Learning on Hybrid Data

**Qinbin Li**
UC Berkeley
qinbin@berkeley.edu

**Chulin Xie**
UIUC
chulinx2@illinois.edu

**Xiaojun Xu**
UIUC
xiaojun3@illinois.edu

**Xiaoyuan Liu**
UC Berkeley
xiaoyuanliu@berkeley.edu

**Ce Zhang**
Together AI, University of Chicago
cez@uchicago.edu

**Bo Li**
UIUC, University of Chicago
bol@uchicago.edu

**Bingsheng He**
National University of Singapore
hebs@comp.nus.edu.sg

**Dawn Song**
UC Berkeley
dawnsong@berkeley.edu

## Abstract

Federated learning has emerged as a promising distributed learning paradigm that facilitates collaborative learning among multiple parties without transferring raw data. However, most existing federated learning studies focus on either horizontal or vertical data settings, where the data of different parties are assumed to be from the same feature or sample space. In practice, a common scenario is the hybrid data setting, where data from different parties may differ both in the features and samples. To address this, we propose HybridTree, a novel federated learning approach that enables federated tree learning on hybrid data. We observe the existence of consistent split rules in trees. With the help of these split rules, we theoretically show that the knowledge of parties can be incorporated into the lower layers of a tree. Based on our theoretical analysis, we propose a layer-level solution that does not need frequent communication traffic to train a tree. Our experiments demonstrate that HybridTree can achieve comparable accuracy to the centralized setting with low computational and communication overhead. HybridTree can achieve up to 8 times speedup compared with the other baselines.

## 1 Introduction

While machine learning models benefit from large training data, data are usually distributed among multiple parties and cannot be transferred due to privacy concerns. Federated Learning (FL) (McMahan et al., 2016; Kairouz et al., 2019; Yang et al., 2019) has been a popular direction to address the above challenge. Existing FL studies mainly focus on horizontal or vertical FL settings. In horizontal FL (HFL), the data of each party shares the same feature space but different sample spaces (e.g., keyboard input behavior of different users). In vertical FL (VFL), the data of each party shares the same sample space but different feature spaces (e.g., data of bank and insurance company on the same user group).

In practical scenarios, hybrid FL is quite common, yet it has not been extensively explored in the current literature. To illustrate this, let's consider a payment network system provider like SWIFT aiming to train a model for detecting anomalous transactions. In this case, the provider can collaborate with multiple banks, which can contribute user-related features for each transaction. Consequently, the data involved in this setting exhibits a hybrid FL configuration, where the data between the

payment network system and the banks originate from different feature spaces, and the data among different banks stem from distinct sample spaces. The hybrid FL setting is particularly prevalent in real-world applications, especially when dealing with tabular data. Each participating party only possesses partial instances or features of the overall global data, leading to the need for effective strategies to leverage such hybrid data for collaborative learning.

On the other hand, the Gradient Boosting Decision Tree (GBDT) is a powerful model, especially for tabular data, which has won many awards in machine learning and data mining competitions (Chen & Guestrin, 2016; Ke et al., 2017). There have been some studies (Cheng et al., 2019; Tian et al., 2020; Li et al., 2023) that design federated GBDT algorithms in the horizontal or vertical FL setting. However, none of the studies work on the hybrid FL setting. They aggregate the information of all parties when training each tree node, and the aggregation strategy relies on the consistency of sample or feature space between different local data. Moreover, high communication and computation overhead are introduced in these node-level solutions. In the presence of a hybrid data setting, it is challenging to design a knowledge aggregation mechanism efficiently and effectively.

To solve the above challenge, we provide key insight for federated tree training with our theoretical analysis: parties contribute simple and neat knowledge to FL which are formulated as split rules (*meta-rule*), and these rules can be incorporated at once in each round. Based on the insight, instead of using node-level solutions that introduce complicated aggregation mechanisms with cryptographic techniques, we design a novel layer-level solution named HybridTree. HybridTree integrates party-specific knowledge by appending layers to the tree structure. Our experiments show that HybridTree can achieve comparable accuracy compared with centralized training while achieving up to eight times speedup compared with node-level solutions.

Our work has the following main contributions.

- We observe the existence of meta-rules in trees. Based on the observation, we propose a tree transformation technique to enable the reordering of split points without compromising the model performance, which supports using only specific features for splitting in the last layer.

- Motivated by the effectiveness of our tree transformation, we propose a new federated tree algorithm on hybrid data, which adopts a novel layer-level tree training strategy that incorporates the parties' knowledge by appending layers.

- We conduct extensive experiments on simulated and natural hybrid federated datasets. Our experiments show that HybridTree is much more efficient than the other baselines with a close accuracy to centralized training.

## 2 BACKGROUND AND RELATED WORK

### 2.1 GRADIENT BOOSTING DECISION TREE

GBDT is a popular model which shows superior performance in machine learning competitions (Chen & Guestrin, 2016; Ke et al., 2017) and real-world applications (Richardson et al., 2007; Kim et al., 2009). It usually achieves better model performance than neural networks for tabular data (McElfresh et al., 2023). The GBDT model contains multiple decision trees. Each tree has two types of nodes: internal nodes that split the input into left or right with a split condition and leaf nodes that output the prediction values. Given an input instance, the final prediction value is computed by summing the prediction values of all trees.

The training of GBDT is a deterministic process. In each iteration, a new tree is trained to fit the residual between the prediction and the target. Formally, given a loss function $\ell$ and a dataset $\mathcal{D} = \{(\mathbf{x}_i, y_i)\}_{i=1}^{n}$, GBDT minimizes the following objective function

$$\mathcal{L} = \sum_i l(y_i, \hat{y}_i) + \sum_k \Omega(\theta_k), \tag{1}$$

where $\hat{y}_i$ is the prediction value, $\Omega(\cdot)$ is a regularization term and $\theta_k$ denotes the parameter of $k$-th decision tree. For the complete training process of GBDT, please refer to Appendix B.2.

## 2.2 FEDERATED GBDT

There have been some federated GBDT algorithms (Li et al., 2023; Tian et al., 2020; Cheng et al., 2019; Zhao et al., 2018; Li et al., 2020; Fang et al., 2021; Wu et al., 2020; Wang et al., 2022; Maddock et al., 2022) for horizontal or vertical FL. Most existing studies (Li et al., 2023; Tian et al., 2020; Cheng et al., 2019; Fang et al., 2021; Wu et al., 2020; Wang et al., 2022; Maddock et al., 2022) adopt a node-level solution that merges the knowledge, which is usually represented by histograms, of different parties when training each node. Different techniques such as homomorphic encryption, secure multi-party computation, and differential privacy are used to protect the transferred information. The node-level solutions suffer from frequent communication traffic and additional computation overhead especially when using cryptographic techniques for privacy protection. Several studies (Li et al., 2020; Zhao et al., 2018) for horizontal FL adopt tree-level solutions. They transfer trees in each round, i.e., each party locally trains GBDTs and transfers them to the next party for boosting. The tree-level solutions may have severe accuracy loss since only local data is used when training each tree. Also, they are not applicable in vertical FL setting since some parties do not have labels to train the local trees. Moreover, all existing federated GBDT studies do not investigate the hybrid FL setting. We have summarized existing federated GBDT studies in Appendix D.

## 2.3 FEDERATED LEARNING ON HYBRID DATA

FL on hybrid data is rarely exploited in the current literature. Zhang et al. (2020) propose to train a feature extractor for every feature in clients, and the server aggregates the feature extractors by feature correspondingly. Such a feature-level aggregation may incur huge computation and memory overhead when the dimension is high. Liu et al. (2020) apply transfer learning in a two-party setting. Two parties locally train the neural networks and a mapping function is used to associate the local outputs and the labels. Both studies are designed for neural networks and are not applicable to trees.

## 3 MOTIVATION AND THEORETICAL SUPPORT

### 3.1 PROBLEM STATEMENT

In this paper, we consider a hybrid FL setting where multiple parties jointly train a GBDT model without transferring data. For ease of presentation, we call the parties with the labels as *hosts* and the parties without labels as *guests*. For simplicity, we start from a scenario involving a single host with multiple guests. Specifically, we assume that a host seeks the help of $N$ guests who have additional features of samples in the host for FL (e.g., a payment system seeks the help of banks for fraud detection). We use $\mathcal{D}_h = \{(\mathbf{x}, y) | \mathbf{x} \in \mathbb{R}^{d_h}\}$ to denote the data of the host and $\mathcal{D}_{g_i} = \{\mathbf{x} | \mathbf{x} \in \mathbb{R}^{d_g}\}$ to denote the data of guest $i$. We use $\mathbf{I}$ to denote the instance ID set of the host and $\mathbf{I}^i$ to denote the instance ID set of guest $i$ ($\mathbf{I} = \cup_{i=1}^{N} \mathbf{I}_i$). Like existing VFL studies (Cheng et al., 2019; Vepakomma et al., 2018), we assume that the data of the host and guests have already been linked (i.e., the host knows whether a guest has additional features for an instance in its local data), which can be achieved by matching anonymous IDs or privacy-preserving record linkage (Gkoulalas-Divanis et al., 2021).

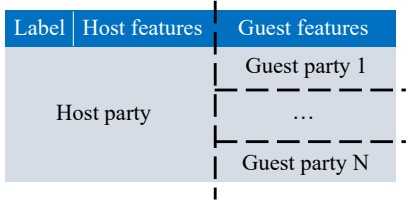

Figure 1: Hybrid data partitioning.

### 3.2 META-RULE AND TREE TRANSFORMATION

As we mentioned in Section 2.2, most existing federated GBDT studies try to aggregate the statistics (e.g., histograms) of each party to update a tree node. When it comes to hybrid FL, one may design a complicated framework that utilizes cryptographic techniques to aggregate the statistics, which would incur large computation and communication overhead. However, *is it necessary to use statistics of all parties to update every node?* Next, to answer this question, we present a key insight: *meta-rules* widely exists in GBDTs. Then, we show that we can transform trees to enable layer-level tree updating based on the meta-rules.

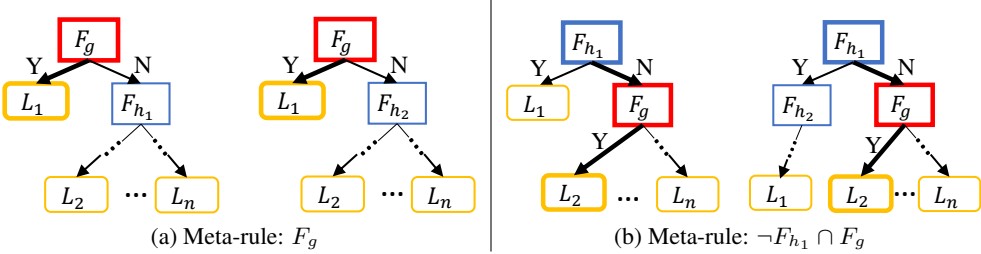

(a) Meta-rule: $F_g$  (b) Meta-rule: $\neg F_{h_1} \cap F_g$

Figure 2: Two examples of meta-rules. $F$ is the split condition and $L$ is the leaf value. In (a), $F_g \rightarrow L_1$ exists in both trees. In (b), $\neg F_{h_1} \rightarrow F_g \rightarrow L_2$ exists in both trees.

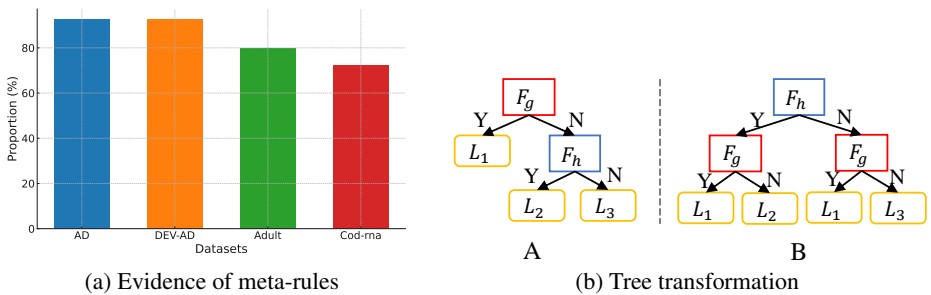

(a) Evidence of meta-rules  (b) Tree transformation

Figure 3: (a) The proportion of trees that have the same meta-rules. (b) $F_g$ is a split rule with the split feature from guests. $F_h$ is a split rule with the split feature from the host. $L$ represents leaf nodes.

**Existence of Meta-Rules**  We start by investigating the properties of GBDTs in hybrid federated datasets. We use two datasets provided by PETs prize challenge (DrivenData). The datasets contain synthetic transaction data provided by a payment network system (host) and account data provided by multiple banks (guests). We train a GBDT model with 50 trees in the centralized setting by linking these datasets without privacy constraints. Analyzing the output model, we focus on split rules (i.e., the joint split condition from the root node to the leaf node) involving features from the guests. Interestingly, for these split rules, we observe that the same rule consistently appear in over 90% of the trees. We present two examples in Figure 2. In Figure 2a, the split rule $F_g$ exists in both trees, i.e., the prediction value is deterministic if $F_g$ is true. In Figure 2b, the split rule $\neg F_{h_1} \cap F_g$ exists in both trees, i.e., the prediction value is deterministic if $\neg F_{h_1} \cap F_g$ is true. As long as it satisfies the split rule, the prediction value is independent of other features. For the sake of clarity, we define such split rules as *meta-rules*.

**Definition 1. (Meta-Rule)** Given a split rule $S := \cap_{j=1}^{N} F_j$ where $F_j$ is a split condition, we call $S$ as a meta-rule if $P(y|x \in S) = P(y|x \in (S \cap F_k)), \forall F_k \neq F_j (j \in [1, N])$.

In the context of tabular data, it is intuitive that guests often contribute simple and neat knowledge in the form of meta-rules. For example, if banks know that a user account has already been closed, then the transactions made by this closed account have a high probability of being anomalous. If a patient's iWatch records an unstable heart rate in daily life, then the hospital may guess that the patient has a heart disease combined with other measurements. To support our assumption, we use four tabular datasets (details of the datasets are available in Section 5.1) to further verify the popularity of meta-rules. For each dataset, we train a GBDT model with 40 trees in the centralized setting. Figure 3a records the proportion of trees where the same meta-rule that determines the prediction value appears. We can observe that most of the trees have the meta-rules in five datasets. Thus, in hybrid FL, to aggregate the knowledge of participants, we focus on how to incorporate the knowledge defined by these meta-rules during training efficiently and effectively.

**Tree Transformation based on Meta-Rule**  Based on the existence of meta-rules, it is not necessary to consider the statistics of all parties when updating each node. We look at a simple tree with depth 2 as shown in Figure 3b as an example. We have the meta-rule $F_g$, i.e., the prediction is independent

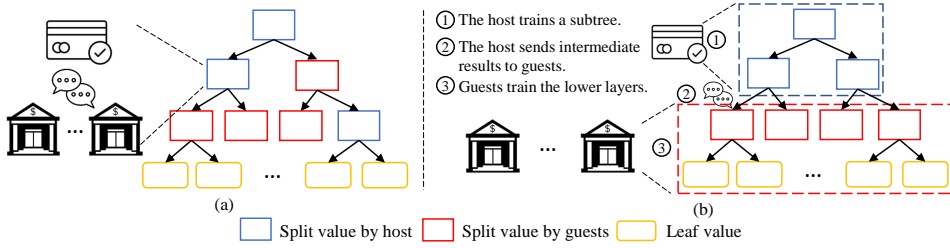

Figure 4: A comparison between node-level solution (a) and our layer-level solution (b). All parties jointly update each node in (a) while each party only updates a segmented tree individually in (b).

of features $F_h$ as long as $F_g$ is true. We can transform Tree A to Tree B by reordering the split node $F_g$ into last layer of the tree. We have the following theorems. The proofs are available in Appendix A of the supplementary material.

**Theorem 2.** *Suppose $F_g$ is a meta-rule in Tree A. For any input instance $\boldsymbol{x} \in \mathcal{D}$, we have $E[f(\boldsymbol{x}; \theta_A)] = E[f(\boldsymbol{x}; \theta_B)]$, i.e., the expectation of prediction value of Tree A and Tree B are the same.*

While the above theorem is based on Figure 3b with a tree of depth two, it can easily be extended to the case with a larger depth of trees by considering $F_g$ as a subtree with split features from guests and $F_h$ as a subtree with split features from hosts. To demonstrate that the split point with guest features can be reordered into the last layers while keeping the model performance, we have the following theorem.

**Theorem 3.** *Suppose $S_m := F_h \cap ... \cap F_g$ is a meta-rule in tree $\theta_A$ where $F_g$ is a split condition using the feature from the guests. For any tree path in tree $\theta_A$ involving the split nodes in $S_m$, we can always reorder the split nodes in the tree path such that $F_g$ is in the last layer. Moreover, naming the tree after the reordering as $\theta_B$, we have $E[f(\boldsymbol{x}; \theta_A)] = E[f(\boldsymbol{x}; \theta_B)]$ for any input instance $\boldsymbol{x} \in \mathcal{D}$.*

From Theorem 3, based on the meta-rule contributed by the guests, we can reorder the split nodes such that the split feature from guests is in the last layers. Thus, it is not necessary to consider all features as possible split values in each tree node as we can incorporate the knowledge by just using features from guests in the last layers. Based on this insight, we propose HybridTree, an efficient and effective hybrid federated GBDT algorithm.

# 4  OUR METHOD: HYBRIDTREE

In this section, inspired and supported by our tree transformation based on meta-rules, we propose the HybridTree approach. In the training, HybridTree adopts a layer-wise training design, where the host party trains a subtree and the guest parties further update the bottom layers. Then, in the inference, as each tree is divided into multiple parties, the host and guest parties collaboratively build the split path of an input instance and make the prediction. Next, we introduce the training and inference processes in detail.

## 4.1  HYBRIDTREE TRAINING

**Overview**  Existing node-level solutions for horizontal or vertical FL require all parties to communicate and jointly update every node as shown in Figure 4(a). Supported by our theoretical analysis, we design a layer-level solution as shown in Figure 4(b), where the host and guests train segmented trees individually without communication during local training. There are three steps in each round. First, the host trains a subtree using its local features and labels. Then, the host sends the encrypted gradients of the instances in the last layer to guests using additively homomorphic encryption (AHE) (Paillier, 1999). Last, guests update the following lower layers of the tree using their local features and receive encrypted gradients, and send back the encrypted prediction values. During each round, the host and guests only communicate twice to incorporate the meta-knowledge from guests, which saves a lot of communication traffic compared with node-level solutions.

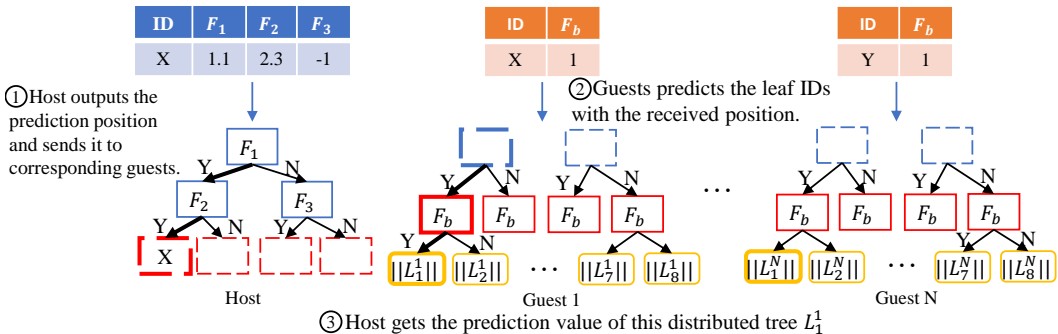

Figure 5: The inference process of HybridTree.

The detailed algorithm is shown in Algorithm 1. Specifically, before training the model, the host initializes the prediction value to zero and generates key pairs for AHE, where the public key is sent to guests (Lines 1-4). For every pair of guests, a common key is generated and exchanged through Diffie-Hellman key exchange (Merkle, 1978), which will be used later for secure aggregation (Bonawitz et al., 2016) (Lines 5-6). In each round, the host updates the gradients of the training data, which is used to train a subtree (Lines 7-9). The $TrainTree()$ algorithm follows the typical GBDT training algorithm, which we present in Appendix B of the supplementary material. Then, for each last-layer node, the host sends the instance ID set and last-layer gradients to guests that have the corresponding instances (Lines 10-13). Note that gradients are computed based on the prediction value and the true label, and raw gradients may leak information about the labels. Thus, we apply AHE to protect the gradients (Line 11), which supports the addition of encrypted values. After receiving the encrypted gradients, guests can compute the leaf values according to Eq. 8 while using the public key to sum encrypted gradients (Lines 16-21). After receiving the encrypted leaf values, the host aggregates and decrypts it using the private key and updates the prediction values (Lines 14-15).

In general, there are three steps in the whole training process: 1) The host party updates a subtree individually (Lines 1-9); 2) The host party sends the encrypted intermediate results into the guest parties (Lines 10-13); 3) The guest parties update the bottom layers individually and send back the encrypted prediction values (Lines 14-21). Since HybridTree does not require accessing all features and instances when updating each node, it can handle the hybrid data case where each party only has partial instances and features. Moreover, based on our analysis in Section 3, by updating the bottom layers using the guests' features, the meta-rule knowledge of the guest parties can be effectively incorporated.

## 4.2 HYBRIDTREE INFERENCE

After HybridTree training, the whole model is distributed among different parties, and collaborative inference is required to predict an input instance like existing vertical FL studies (Cheng et al., 2019). We present the inference process in Figure 5. Still, we assume that the test data among different parties have already been linked by ID before inference. First, the host splits the input instance into a last-layer node using its subtree and sends the position of the predicted node to guests that have the instances. Then, the guests further split the instance with the received position and return the predicted leaf location. Last, the server averaged the prediction values of the received locations to get the final prediction value. During the whole prediction process, only two communication times are needed and all test instances can be processed in parallel.

## 4.3 PRIVACY GUARANTEES

We provide the same privacy guarantee as existing vertical federated learning studies on GBDTs (Li et al., 2023; Cheng et al., 2019). We assume that all the parties are honest-but-curious, where they strictly follow the algorithm and do not collude with each other. During the training process, the host only receives the encrypted prediction values from guests and guests only receive the encrypted gradients from the host. Thus, there is no information leakage in the training. During the inference process, the host and guests only receive the predicted node locations from each other, without

---

**Algorithm 1:** The HybridTree training algorithm

---

**Input:** Host dataset $\mathcal{D}_h = \{(\mathbf{x}_i, y_i)\}_{i=1}^n$ with instance ID set $\mathbf{I}$, guests' datasets $\mathcal{D}_g^i$ $(i \in [N])$, the depth of tree trained by the host $E_h$, the depth of tree trained by guests $E_g$, number of trees $T$, loss function $\ell$, regularization term $\lambda$.

**Output:** The final model $\theta$

```
   /* Conducted on host */
```
1   **HostTrain**$(\mathcal{D}_h, \mathbf{I}, E_h, E_g, T, \ell)$:
2   $\mathbf{y}_p \leftarrow [\mathbf{0}]$             `// Initialize prediction value to zero`
3   $k_{pub}, k_{pri} \leftarrow GenerateKeys()$     `// Generate homomorphic encryption keys`
4   Send $k_{pub}$ to guests
5   **for** every pair of guests $(G_i, G_j)(i \neq j)$ **do**
6     $k_{ij} \leftarrow DHKey()$    `// Generate common key through DH key exchange`
7   **for** $t = 1, 2, ..., T$ **do**
8     $\mathbf{G} \leftarrow [\partial_{y_p^i} \ell(y, y_p^i)]_{i=1}^n$                    `// Update gradients`
9     $\{\mathbf{I}_i\}_{i=1}^k, \{\mathbf{G}_i\}_{i=1}^k \leftarrow TrainTree(\mathbf{I}, \mathbf{G}, E_h)$     `// Train a subtree and get` $k$
      `last-layer nodes`
10     **for** each last-layer node $i$ **in parallel do**
11       $\|\mathbf{G}_i\| \leftarrow Enc(\mathbf{G}_i, k_{pri})$              `// Encrypt gradients`
12       **for** each guest $u$ **in parallel do**
13         Send $\mathbf{I}_i^u, \|\mathbf{G}_i^u\|$ to Guest $u$   `// Send intermediate results to guest`
14         $\|\mathbf{y}_p^u\| \leftarrow GuestTrain(\mathbf{I}_i^u, \|\mathbf{G}_i^u\|, E_g)$     `// Guest updates the bottom`
           `layers`
15     $\mathbf{y}_p \leftarrow \mathbf{y}_p + Dec(\sum_{u \in N} \|\mathbf{y}_p^u\|, k_{pri})$         `// Update prediction values`

```
   /* Conducted on guests */
```
16   **GuestTrain**$(\mathbf{I}, \|\mathbf{G}\|, E_g)$:                    `// Update non-leaf layers`
17   $\{\mathbf{I}_i\}_{i=1}^k, \{\|\mathbf{G}_i\|\}_{i=1}^k \leftarrow TrainTree(\mathbf{I}, \|\mathbf{G}\|, E_g)$
18   **for** each last-layer node $i$ **do**
19     $\|\mathbf{V}_i\| \leftarrow \frac{\sum_j \|\mathbf{G}_i^j\|}{|\mathbf{I}_i| + \lambda}$                `// Compute leaf values`
20     $\|\mathbf{y}_p^{\mathbf{I}_i}\| \leftarrow \|\mathbf{V}_i\| + \sum_j k_{\cdot j} - \sum_j k_{j \cdot}$   `// Add noises for secure aggregation`
21   **return** $y_p$

---

information about the data or model of the other parties. Note that there may be potential inference attacks and techniques like differential privacy (Dwork, 2011) can be applicable (Li et al., 2023), which is out of the scope of the paper.

## 5   EVALUATION

### 5.1   EXPERIMENTAL SETTINGS

**Datasets**   We use four datasets in our experiments: 1) Two versions of hybrid FL datasets provided by PETs Prize Challenge for anomalous transaction detection. In the datasets, one party (i.e., host) holds the synthetic transaction data and the label and multiple parties (i.e., guests) hold the account data. Both datasets have 25 guests. 2) Two simulated hybrid federated learning datasets. We generate these two datasets by partitioning the centralized tabular datasets Adult and Cod-rna into multiple subsets randomly. We first divide the dataset vertically to get a host dataset and then divide the remaining one horizontally to get multiple guest datasets. The number of guest parties is set to 5 for both datasets. For more details about the datasets, please refer to Appendix C. In the experiments of the main paper, all guests share the same feature spaces and different sample spaces. For results on more experimental settings, please refer to Appendix C.

**Approaches**   We compare the following approaches with HybridTree: 1) **ALL-IN**: We train a GBDT model on the global data without any privacy constraints. This approach represents the upper

Table 1: The comparison of model performance between different approaches. For FedTree, Secure-Boost, and Pivot, we run them with every possible guest and report the minimum and maximum model performance achieved.

|         | HybridTree | SOLO  | FedTree     | SecureBoost | Pivot       | TFL   | ALL-IN |
|---------|------------|-------|-------------|-------------|-------------|-------|--------|
| AD      | **0.689**  | 0.492 | 0.537-0.566 | 0.537-0.566 | 0.534-0.561 | 0.530 | 0.703  |
| DEV-AD  | **0.553**  | 0.111 | 0.412-0.462 | 0.412-0.462 | 0.414-0.468 | 0.397 | 0.574  |
| Adult   | **0.832**  | 0.653 | 0.764-0.788 | 0.764-0.788 | 0.755-0.778 | 0.773 | 0.853  |
| Cod-rna | **0.927**  | 0.690 | 0.805-0.863 | 0.805-0.863 | 0.811-0.870 | 0.884 | 0.931  |

bound of model performance. 2) **SOLO**: The host locally trains a GBDT model with its local data. This approach represents the lower bound of model performance. 3) **2-party VFL**: The host party collaborates with one of the guests to conduct vertical federated GBDT. We compare three vertical federated learning studies for GBDTs, including **FedTree** (Li et al., 2023), **SecureBoost** (Cheng et al., 2019), and **Pivot** (Wu et al., 2020). We run each approach with every possible guest and report the minimum and maximum model performance achieved. Note that it is non-trivial to apply the VFL studies to the hybrid data setting with multiple guest parties. 4) **TFL**: We assume that guests have the labels and adopt a tree-level solution (Zhao et al., 2018; Li et al., 2020) with all parties, i.e., each party trains a tree individually and sequentially. We use this approach to assess the effectiveness of tree-level knowledge aggregation.

**Model and Metrics** We train a GBDT model with 50 trees. The learning rate is set to 0.1. The maximum depth is set to 7 for the baselines. The maximum depth for the host is set to 5 and the maximum depth for guests is set to 2 for HybridTree so that the total depth of the tree is 7 to ensure a fair comparison. The regularization term $\lambda$ is set to 1. For AD and DEV-AD, we use AUPRC (Area Under Precision-Recall Curve) as the metric since these two datasets are highly class-imbalanced. For two simulated datasets, we use classification accuracy as the metric.

We run experiments on a machine with four Intel Xeon Gold 6226R 16-Core CPUs. We fix the number of threads to 10 for each experiment. Due to the page limit, we only present some of the results in the main paper. For more experimental results, please refer to Appendix C of the supplementary material.

## 5.2 MODEL PERFORMANCE

We compare the model performance of HybridTree with the other baselines with the results exhibited in Table 1. Given the deterministic nature of the GBDT training process, the output remains consistent across multiple runs, rendering the reporting of mean and standard deviation unnecessary. The results reveal that HybridTree's performance closely mirrors that of ALL-IN, which represents the upper-bound performance without privacy restrictions. Furthermore, HybridTree consistently surpasses the model performance of SOLO, FedTree, Pivot, and TFL by a substantial margin. FedTree and Pivot, which relies solely on data from a single guest for training, suffers a significant accuracy deficit. TFL, on the other hand, adopts a tree-level knowledge aggregation strategy, which falls short in effectiveness since each tree is inherently weak. In contrast, our method skillfully amalgamates the knowledge of guests utilizing a layer-level design.

## 5.3 TRAINING PERFORMANCE

We contrast the communication and computational efficiency during training of HybridTree and VFL approaches in Table 2. The comparison of inference performance is presented in Appendix C of the supplementary material. We limit our comparison to VFL approaches, as other methodologies break the privacy constraints by sharing data/labels and do not impose additional communication or computational burdens. As the table illustrates, HybridTree significantly outperforms FedTree and Pivot in both communication costs and training duration. The communication speed can be accelerated up to six times, while the computational speed may see an enhancement of up to eight times. HybridTree primarily incorporates lightweight AHE for encryption. These encrypted gradients are transmitted only once per tree. Furthermore, cryptographic operations are restricted to the lower

Table 2: The training efficiency comparison between HybridTree and VFL approaches. The speedup of HybridTree is computed by comparing FedTree.

| | Communication size (GB) | | | | | Training time (s) | | | | |
|---|---|---|---|---|---|---|---|---|---|---|
| | HybridTree | FedTree | SecureBoost | Pivot | speedup | HybridTree | FedTree | SecureBoost | Pivot | speedup |
| AD | 223.6 | 1363.9 | 1389.2 | 1420.3 | **6.1x** | 84.1 | 595.6 | 3212.7 | 316823 | **7.1x** |
| DEV-AD | 142.6 | 770.1 | 681.9 | 792.2 | **5.4x** | 58.2 | 464.9 | 2856.6 | 284235 | **8.0x** |
| Adult | 1.55 | 9.74 | 14.6 | 11.9 | **6.3x** | 2.0 | 8.6 | 71.1 | 9234 | **4.3x** |
| Cod-rna | 2.84 | 15.92 | 20.4 | 18.5 | **5.6x** | 1.0 | 5.3 | 24.3 | 3845 | **5.3x** |

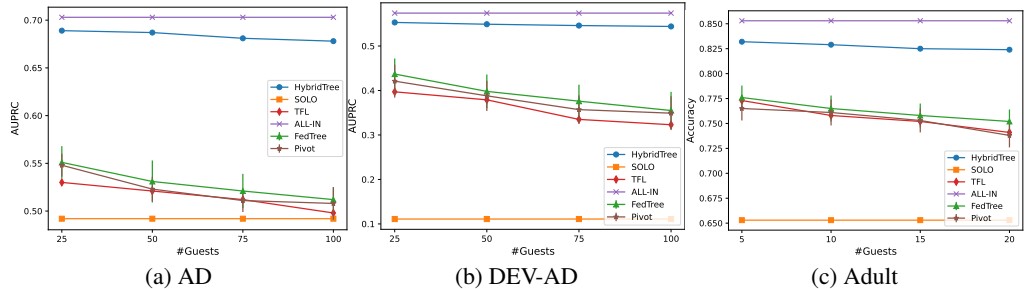

| (a) AD | (b) DEV-AD | (c) Adult |
|---|---|---|

Figure 6: Model performance of different approaches by varying the number of guests. We omit SecureBoost as its curve overlaps with FedTree.

Table 3: Model performance of different approaches in the multi-host setting.

| | HybridTree | SOLO | FedTree | SecureBoost | Pivot | TFL | ALL-IN |
|---|---|---|---|---|---|---|---|
| AD | **0.682** | 0.423-0.443 | 0.498-0.502 | 0.498-0.504 | 0.483-0.492 | 0.512 | 0.703 |
| DEV-AD | **0.548** | 0.094-0.099 | 0.389-0.425 | 0.387-0.423 | 0.392-0.438 | 0.366 | 0.574 |
| Adult | **0.828** | 0.582-0.591 | 0.712-0.722 | 0.712-0.722 | 0.698-0.710 | 0.730 | 0.853 |
| Cod-rna | **0.911** | 0.621-0.635 | 0.784-0.804 | 0.784-0.804 | 0.771-0.792 | 0.821 | 0.931 |

layers in HybridTree, in contrast to node-level solutions where they occur at every node. As a result, HybridTree demonstrates superior efficiency compared to the baselines.

## 5.4 SCALABILITY

We manipulate the number of guests from 25 to 100 for AD and DEV-AD, and from 5 to 20 for Adult and Cod-rna by randomly dividing each guest dataset into multiple subsets. The corresponding results are presented in Figure 6. Due to the page limit, we leave the results of Cod-rna in Appendix C of the supplementary material. From the results, it is evident that HybridTree exhibits significantly more stability than FedTree, Pivot, and TFL. Even when the number of guests is increased, HybridTree can well consolidate the knowledge from all parties. In contrast, FedTree, Pivot, and TFL exhibit a considerable degradation in performance when local knowledge is limited.

## 6 CONCLUSIONS

This paper introduces HybridTree, a new federated GBDT algorithm designed for a hybrid data environment. Leveraging our insights into meta-rules, we propose a tree transformation capable of reordering split features. Building upon this transformation, we introduce an innovative hybrid tree learning algorithm that integrates the knowledge of guests by directly appending layers. Experimental results demonstrate that HybridTree significantly outperforms other baseline methodologies in terms of efficiency and effectiveness. While HybridTree is designed for GBDT due to its popularity, the idea of layer-level training is applicable to other trees. We consider hybrid federated learning on multi-modal data as future work.

ACKNOWLEDGEMENTS

This research is supported by the National Research Foundation Singapore and DSO National Laboratories under the AI Singapore Programme (AISG Award No: AISG2-RP-2020-018), Singapore National Research Foundation funding #053424, ARL funding #W911NF-23-2-0137, DARPA funding #112774-19499, IC3 industry partners, the National Science Foundation under grant no. 2229876 and funds provided by the National Science Foundation, by the Department of Homeland Security, by IBM, and by JPMorgan Chase & Co. Any opinions, findings and conclusions or recommendations expressed in this material are those of the authors and do not reflect the views of the supporting entities.

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

APPENDICES

In Appendix A, we prove the theorems introduced in Section 3. In Appendix B, we introduce the algorithmic details of training a tree. In Appendix C, we present additional experimental details and results. In Appendix D, we discuss the potential broader impacts and limitations of our approach.

## A   PROOF

**Definition 1.** (**Meta-Rule**) Given a split rule $S := \cap_{j=1}^{N} F_j$ where $F_j$ is a split condition by feature $f_j$, we call it as meta-rule if $P(y|\mathbf{x} \in S) = P(y|\mathbf{x} \in (S \cap F_k))$ for any $F_k$ not in split rule $S$.

**Theorem 2.** *Suppose $F_g$ is a meta-rule in Tree A. For any input instance $\boldsymbol{x} \in \mathcal{D}$, we have $E[f(\boldsymbol{x}; \theta_A)] = E[f(\boldsymbol{x}; \theta_B)]$, i.e., the expectation of prediction value of Tree A and Tree B are the same.*

*Proof.* For ease of presentation, we use $\{F\}$ to denote the instance ID set that satisfies split rule $F$ (i.e., $\{F\} := \{i|\mathbf{x}_i \in F\}$). We consider the instances sets of three leaf nodes in Tree A.

1) For the instance set $\{F_g\}$ in $L_1$ of Tree A, it will be divided into two sets in Tree B: $\{F_h \cap F_g\}$ in $L_1'$ and $\{\neg F_h \cap F_g\}$ in $L_3'$. The expectation of leaf value $L_1'$ is

$$E(L_1') = -\frac{E(\sum_{i \in \{F_h \cap F_g\}} g_i)}{|\{F_h \cap F_g\}|} \tag{2}$$

From Definition 1, we have $P(y|\mathbf{x} \in F_g) = P(y|\mathbf{x} \in (F_h \cap F_g))$. Note that gradient $g$ is a mapping from $y$. We have $P(g|\mathbf{x} \in F_g) = P(g|\mathbf{x} \in (F_h \cap F_g))$. Thus, we have

$$\begin{aligned} E(L_1') &= -\frac{|\{F_h \cap F_g\}|}{|F_g|} \cdot \frac{E(\sum_{i \in \{F_g\}} g_i)}{|\{F_h \cap F_g\}|} \\ &= -\frac{E(\sum_{i \in \{F_g\}} g_i)}{|F_g|} \\ &= E(L_1). \end{aligned} \tag{3}$$

Similarly, we have

$$\begin{aligned} E(L_3') &= -\frac{|\{\neg F_h \cap F_g\}|}{|F_g|} \cdot \frac{E(\sum_{i \in \{F_g\}} g_i)}{|\{\neg F_h \cap F_g\}|} \\ &= -\frac{E(\sum_{i \in \{F_g\}} g_i)}{|F_g|} \\ &= E(L_1). \end{aligned} \tag{4}$$

2) For the instance set $\{\neg F_g \cap F_h\}$ in $L_2$ of Tree A, it will be relocated to $L_2'$ of Tree B.

3) For the instance set $\{\neg F_g \cap \neg F_h\}$ in $L_3$ of Tree A, it will be relocated to $L_4'$ of Tree B.

Thus, for any instance $x$, $E[f(x; \theta_A)] = E[f(x; \theta_B)]$.  □

**Theorem 3.** *Suppose $S_m := F_h \cap ... \cap F_g$ is a meta-rule in tree $\theta_A$ where $F_g$ is a split condition using the feature from the guests. For any tree path in tree $\theta_A$ involving the split nodes in $S_m$, we can always reorder the split nodes in the tree path such that $F_g$ is in the last layer. Moreover, naming the tree after the reordering as $\theta_B$, we have $E[f(\boldsymbol{x}; \theta_A)] = E[f(\boldsymbol{x}; \theta_B)]$ for any input instance $\boldsymbol{x} \in \mathcal{D}$.*

*Proof.* We use $\theta_g$ to denote the subtree with root node $F_g$. For ease of presentation, we use $\mathbf{F}_h \cap F_g$ to denote the given meta-rule, where $\mathbf{F}_h$ is the split rule with split features from the host. Without loss of generality, we assume that the left child node of $F_g$ is a leaf node when $F_g$ is true, denoted as $L_l$. For every possible partial split rule in the subtree of the right child node $\mathbf{F}_k := \cap_j F_j$, we have

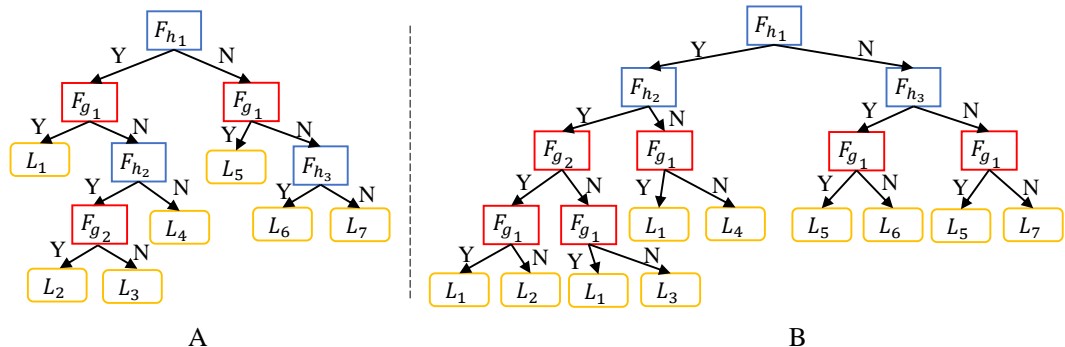

Figure 7: Suppose $F_{h_1} \cap F_{g_1}$ is a meta-rule. Tree A can be transformed into Tree B.

$P(g|\mathbf{x} \in \mathbf{F}_h \cap F_g) = P(g|\mathbf{x} \in \mathbf{F}_h \cap F_g \cap \mathbf{F}_k)$. By moving $F_g$ to the last layer of right child node, for each split rule $\mathbf{F}_h \cap \mathbf{F}_k$, it generates two new leaf nodes $L'_l$ ($\mathbf{F}_h \cap \mathbf{F}_k \cap F_g$) and $L'_r$ ($\mathbf{F}_h \cap \mathbf{F}_k \cap \neg F_g$). We have

$$
\begin{aligned}
E(L'_l) &= -\frac{E(\sum_{i\in\{\mathbf{F}_h\cap\mathbf{F}_k\cap F_g\}} g_i)}{|\{\mathbf{F}_h \cap \mathbf{F}_k \cap F_g\}|} \\
&= -\frac{|\{\mathbf{F}_h \cap \mathbf{F}_k \cap F_g\}|}{|\{\mathbf{F}_h \cap \mathbf{F}_k\}|} \cdot \frac{E(\sum_{i\in\{\mathbf{F}_h\cap\mathbf{F}_k\}} g_i)}{|\{\mathbf{F}_h \cap \mathbf{F}_k \cap F_g\}|} \\
&= -\frac{E(\sum_{i\in\{\mathbf{F}_h\cap\mathbf{F}_k\}} g_i)}{|\{\mathbf{F}_h \cap \mathbf{F}_k\}|} \\
&= E(L_l)
\end{aligned}
\tag{5}
$$

For $L'_r$, it is equivalent to the original tree node $\neg F_g \cap \mathbf{F}_h \cap \mathbf{F}_k$.

Thus, the expectation of the prediction value remains unchanged for any input instance through our transformation. □

Based on Theorem 3, we can transform a tree by reordering the split points such that the split points using the guest features are in the bottom layers. Figure 7 shows an example.

## B  NOTATIONS AND ALGORITHM

### B.1  NOTATIONS

The notations used in the paper are summarized in Table B.1.

### B.2  THE GBDT TRAINING ALGORITHM

At the $t$-th iteration using second-order approximation (Si et al., 2017), GBDT minimizes the following objective function

$$
\begin{aligned}
\tilde{\mathcal{L}}^{(t)} &= \sum_i l(y_i, \hat{y}_i^{t-1} + f_t(\mathbf{x}_i; \theta_t)) + \Omega(\theta_t) \\
&\approx \sum_i [l(y_i, \hat{y}_i^{t-1}) + g_i f_t(\mathbf{x}_i; \theta_t) + \frac{1}{2} f_t^2(\mathbf{x}_i; \theta_t)] + \Omega(\theta_t)
\end{aligned}
\tag{6}
$$

where $g_i = \partial_{\hat{y}^{(t-1)}} l(y_i, \hat{y}^{(t-1)})$ is first order gradient on the loss function and $f_t(\cdot)$ is the tree function.

GBDT updates a tree from the root node to minimize Eq (6) until reaching the specified maximum depth. We use $\mathbf{I}$ to denote the instance ID set in the current node. If the current node is a split node,

Table 4: Notations used in the paper.

| Notation | Decsription |
|---|---|
| $\mathcal{D}_h$ | Dataset of the host party |
| $\mathcal{D}_g^i$ | Dataset of guest party $i$ |
| $\mathcal{I}$ | Instance ID set |
| $E_h$ | The depth of tree trained by the host party |
| $E_g$ | The depth of tree trained by the guest party |
| $T$ | Number of trees |
| $\ell$ | Loss function |
| $\lambda$ | Hyperparameter |
| $\mathbf{y}_p$ | Prediction value vector |
| $k_{pub}$ | Publick key of homomorphic encryption |
| $k_{pri}$ | Private key of homomorphic encryption |
| $G_i$ | Guest party $i$ |
| $k_{ij}$ | Key for guest pair $(G_i, G_j)$ generated by DH key exchange |
| $\mathbf{G}$ | First-order gradients in GBDTs |
| $\mathbf{V}$ | Leaf values |
| $U$ | Gain of a split |

suppose the split value splits $\mathbf{I}$ into $\mathbf{I}_L$ and $\mathbf{I}_R$. Then, the gain of the split value is defined by the loss reduction after split, which is

$$U = \frac{(\sum_{i \in \mathbf{I}_L} g_i)^2}{|\mathbf{I}_L| + \lambda} + \frac{(\sum_{i \in \mathbf{I}_R} g_i)^2}{|\mathbf{I}_R| + \lambda}. \tag{7}$$

Since it would be computationally expensive to traverse all possible split values to find the one with the maximum gain, GBDT usually considers a small number of cut points as possible split candidates. The best split point is selected from these split candidates. If the tree reaches the maximum depth or if the gain remains negative, the current node becomes a leaf node. To minimize Eq (6), the optimal leaf value is

$$V = -\frac{\sum_{i \in \mathbf{I}} g_i}{|\mathbf{I}| + \lambda} \tag{8}$$

After training a tree according to Eq. (7) and Eq. (8), we can update the gradients using the current prediction value and train the next tree until reaching the specified number of trees.

The algorithm for training a tree in GBDT, denoted as $TrainTree()$, is presented in Algorithm 2. When the maximum depth is reached, the leaf value is computed based on the gradients (Lines 2-4). On the other hand, if the maximum depth is not reached, the algorithm proceeds to calculate the gain for each potential split value (Lines 6-17) and stores the split value with the highest gain. If the gain is greater than zero, the instances are split using the recorded split value, and two subtrees are trained as separate branches (Lines 18-20). However, if the gain is not greater than zero, the current node does not require further splitting and is treated as a leaf node (Lines 21-23).

---

**Algorithm 2:** Train a single tree in GBDT.

---

**Input:** Instance ID set $\mathbf{I}$, gradients $\mathbf{G}$, maximum depth $E$.
**Output:** The final model $\theta$

1   **TrainTree**($\mathbf{I}, \mathbf{G}, E$):
2   **if** $E == 1$ **then**
3     $V \leftarrow -\frac{\sum_{i \in \mathbf{I}} g_i}{|\mathbf{I}|}$                 // Compute leaf value
4     set the current node to a leaf node with value $V$

5   **else**
6     $S_{max} \leftarrow 0$
7     **for** every possible split rule $F_j$ **do**
8       $\mathbf{I}_l \leftarrow \{i | x_i \in F_j\}$
9       $\mathbf{I}_r \leftarrow \{i | x_i \notin F_j\}$
10      $S_l \leftarrow \frac{(\sum_{i \in \mathbf{I}_l} g_i)^2}{|\mathbf{I}_l|}$
11      $S_r \leftarrow \frac{(\sum_{i \in \mathbf{I}_r} g_i)^2}{|\mathbf{I}_r|}$
12      $S \leftarrow S_l + S_r$               // Compute gain
13      **if** $S > S_{max}$ **then**
14        set the current node to a split node with rule $F_j$
15        $S_{max} \leftarrow S$
16        $\mathbf{I}_L \leftarrow \mathbf{I}_l$
17        $\mathbf{I}_R \leftarrow \mathbf{I}_r$

18     **if** $G_{max} > 0$ **then**
19      $TrainTree(\mathbf{I_L}, \mathbf{G_{I_L}}, E-1)$       // Train a subtree recursively
20      $TrainTree(\mathbf{I_R}, \mathbf{G_{I_R}}, E-1)$
21     **else**
22      $V \leftarrow -\frac{\sum_{i \in \mathbf{I}} g_i}{|\mathbf{I}|}$
23      set the current node to a leaf node with value $V$

---

Table 5: Statistics of the datasets.

|        | #training instances | #test instances | #features of host | #features of guests | #guests |
|--------|---------------------|-----------------|-------------------|---------------------|---------|
| AD     | 4,691,615           | 705,108         | 9                 | 4                   | 25      |
| DEV-AD | 2,993,804           | 1,003,675       | 9                 | 4                   | 25      |
| Adult  | 32,561              | 16,281          | 102               | 21                  | 5       |
| Cod-rna| 44,651              | 14,884          | 6                 | 2                   | 5       |

## C  EXPERIMENTS

### C.1  DATASETS

The dataset statistics are presented in Table 5. To create the host dataset for Adult and Cod-rna[1], we employ a random sampling approach. Specifically, we generate a random number between zero and the total number of features, and assign this sampled number of features to the host dataset. The remaining features are then partitioned randomly and equally into five subsets, resulting in the generation of five guest datasets in the default setting.

### C.2  MULTI-HOST SETTING

While we assume there is only one host in our design for simplicity, HybridTree can be easily extended to the multi-host setting, where multiple hosts (e.g., hospitals) collaborate with guests (patients' wearable health devices) for FL. Each host can follow the HybridTree training process to train a GBDT with the guests that have the corresponding instances of the host. Then, for inference, we can conduct prediction on each GBDT and apply bagging (Breiman, 1996) to aggregate the prediction results of multiple GBDTs. For regression tasks, we average the prediction values of multiple GBDTs as the final prediction value. For classification tasks, we apply max-voting to select the class with the highest voting as the prediction class.

We simulate the multi-host setting by randomly partitioning the host dataset into five subsets. The results of the multi-host settings are shown in Table 3. HybridTree continues to substantially surpass other baseline methods, underscoring the effectiveness of our bagging strategy in a multi-host configuration. Compared to the single-host scenario, the performance of other methodologies markedly deteriorates due to the constrained data availability from the host.

### C.3  HETEROGENEITY

While the hybrid federated datasets naturally have data heterogeneity among different parties, their heterogeneity cannot be easily quantified and controlled. To assess model performance amidst diverse data heterogeneity, we consolidate the guest datasets into a unified global set and divide it into multiple subsets in accordance with the corresponding labels in the host. Specifically, we sample $p_k \sim Dir_{10}(\beta)$ and allocate a $p_{k,j}$ proportion of instances from class $k$ to guest $j$, where $Dir(\beta)$ denotes the Dirichlet distribution with a concentration parameter $\beta$. The heterogeneity intensifies as $\beta$ diminishes. The results for varying $\beta$ values are depicted in Figure 8. HybridTree consistently surpasses other baseline methods across all settings. In the case of VFL, performance is contingent upon the data quality of a single guest, resulting in instability and a substantial error bar.

### C.4  OVERLAPPED SAMPLE AND HETEROGENEOUS FEATURE SETTING

In the experiments conducted in the main paper, all guest datasets share the same feature space, and there are no overlapping samples between the guests. However, it is worth noting that our algorithm does not impose any specific requirements regarding the feature and sample spaces across guests. To simulate this flexible setting, we introduce a simulation where, for each guest dataset, a random number $\alpha$ is generated from the range of $[0, d]$, representing the number of features to be dropped.

---

[1]https://www.csie.ntu.edu.tw/~cjlin/libsvmtools/datasets/

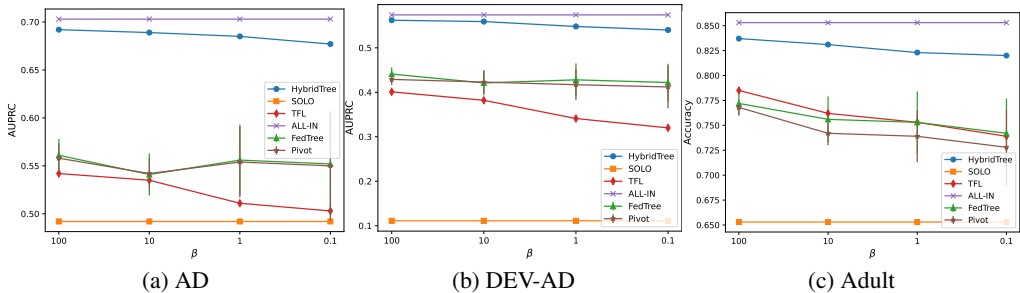

|  |  |  |
|---|---|---|
| (a) AD | (b) DEV-AD | (c) Adult |

Figure 8: Model performance of different approaches by varying heterogeneity. We omit SecureBoost as its curve overlaps with FedTree.

Table 6: The model performance in the setting with overlapped samples and heterogeneous features between different guests.

|  | HybridTree | SOLO | FedTree | SecureBoost | Pivot | TFL | ALL-IN |
|---|---|---|---|---|---|---|---|
| AD | **0.673** | 0.492 | 0.511-0.559 | 0.510-0.557 | 0.515-0.547 | 0.514 | 0.682 |
| DEV-AD | **0.546** | 0.111 | 0.389-0.444 | 0.393-0.443 | 0.372-0.421 | 0.384 | 0.561 |
| Adult | **0.801** | 0.655 | 0.753-0.782 | 0.753-0.782 | 0.759-0.789 | 0.756 | 0.820 |
| Cod-rna | **0.908** | 0.690 | 0.776-0.858 | 0.776-0.858 | 0.771-0.845 | 0.861 | 0.919 |

Additionally, an additional $\beta$ number of samples is assigned from other guest datasets, with $\beta$ drawn randomly from the range of $[0, \frac{n}{20}]$, where $d$ denotes the feature dimension and $n$ represents the total number of samples. The results obtained with this simulated setting are presented in Table 6. HybridTree continues to outperform the other baselines and achieves performance comparable to centralized training, thus showcasing the robustness of HybridTree in various hybrid data settings.

## C.5 OVERHEAD OF HYBRIDTREE

In Table 7, we provide a detailed breakdown of the training time for HybridTree. A comparison with the ALL-IN approach reveals that the primary computational overhead of HybridTree lies in the update process of the last layers in the guest models. This step involves computations on encrypted gradients, which can be computationally expensive. However, thanks to its layer-level design, HybridTree significantly reduces the computation overhead compared to node-level solutions.

## C.6 INFERENCE PERFORMANCE

The inference costs of HybridTree and VFL are presented in Table 8. Since HybridTree and VFL approaches have a similar inference procedure, both approaches have a low inference time. In VFL approaches, since each tree node may be distributed in host or guests, multiple communication rounds

Table 7: Training overhead of HybridTree compared with ALL-IN.

|  | HybridTree | | ALL-IN |
|---|---|---|---|
|  | Host training time (s) | Guest training time (s) | training time (s) |
| AD | 37.4 | 45.7 | 39.8 |
| DEV-AD | 28.9 | 29.3 | 31.7 |
| Adult | 1.1 | 0.9 | 1.1 |
| Cod-rna | 0.6 | 0.4 | 0.6 |

Table 8: Communication size (MB) and inference time (s) of HybridTree and VFL during prediction. The speedup of HybridTree is computed by comparing to FedTree.

| | Communication size (MB) | | | | | Inference time (s) | | | | |
|---|---|---|---|---|---|---|---|---|---|---|
| | HybridTree | FedTree | SecureBoost | Pivot | speedup | HybridTree | FedTree | SecureBoost | Pivot | speedup |
| AD | 5.6 | 16.4 | 20.5 | 19.2 | **2.8x** | 17.9 | 22.1 | 28.4 | 12528 | **1.2x** |
| DEV-AD | 8.1 | 20.6 | 25.2 | 31.5 | **2.5x** | 25.1 | 28.9 | 34.5 | 9825 | **1.2x** |
| Adult | 0.28 | 1.36 | 1.93 | 2.51 | **4.8x** | 0.92 | 1.35 | 2.09 | 426 | **1.5x** |
| Cod-rna | 0.48 | 1.64 | 1.92 | 2.84 | **3.4x** | 0.89 | 1.37 | 2.28 | 498 | **1.5x** |

Table 9: The comparison of model performance between different approaches. For FedTree, Secure-Boost, and Pivot, we run them with every possible guest and report the minimum and maximum model performance achieved.

| | HybridTree | SOLO | FedTree | SecureBoost | Pivot | TFL | ALL-IN |
|---|---|---|---|---|---|---|---|
| 4 | **0.671** | 0.402 | 0.470-0.476 | 0.470-0.476 | 0.458-0.462 | 0.530 | 0.703 |
| 6 | **0.682** | 0.423 | 0.498-0.502 | 0.498-0.502 | 0.496-0.499 | 0.397 | 0.574 |
| 8 | **0.689** | 0.431 | 0.506-0.511 | 0.506-0.511 | 0.498-0.502 | 0.773 | 0.853 |

may be required during an inference path if it involves changes in node locations between host and guests. In HybridTree, since a tree is divided into two parts, only two communication rounds are required. Thus, HybridTree has a lower communication overhead than FedTree and Pivot.

## C.7 SENSITIVITY STUDY

We change the tree depth from 4 to 8 on AD and present the results in Table 9. While increasing tree depth can increase the model performance, our approach consistently performs better than the other baselines.

## C.8 VERTICAL FEDERATED LEARNING

Our approach is also applicable in the FFL setting, where the number of hosts and guests is exactly one. By merging all guests as a single guest, we compare HybridTree with other VFL studies and the results are shown in Table 10 and Table 11. HybridTree can achieve comparable model performance with VFL studies while significantly reducing the training time.

## C.9 IMPACT OF THE HOST DATASET

To investigate the impact of changes in the host dataset, we use the synthetic hybrid FL dataset Adult. Specifically, in the host dataset, we randomly sample 20-100% instances/features to use in training. The results are shown in Table. While reducing the instances/features will reduce the overall performance of all approaches, HybridTree consistently outperforms the other baselines.

Table 10: The model performance of different approaches in the VFL setting.

| | HybridTree | FedTree | SecureBoost | Pivot |
|---|---|---|---|---|
| AD | 0.702 | 0.708 | 0.708 | 0.704 |
| DEV-AD | 0.594 | 0.603 | 0.603 | 0.597 |
| Adult | 0.851 | 0.862 | 0.862 | 0.858 |
| Cod-rna | 0.944 | 0.957 | 0.957 | 0.951 |

Table 11: The training time (s) of different approaches in the VFL setting.

|        | HybridTree | FedTree | SecureBoost | Pivot  | speedup |
|--------|-----------|---------|-------------|--------|---------|
| AD     | 103.7     | 782.5   | 4628.4      | 425698 | 7.5     |
| DEV-AD | 67.4      | 623.6   | 3745.9      | 395720 | 9.3     |
| Adult  | 3.2       | 12.8    | 101.8       | 13829  | 4.0     |
| Cod-rna| 1.7       | 8.1     | 82.5        | 6023   | 4.8     |

Table 12: The model performance of different approaches by varying the size of the host dataset.

|            | Proportion | HybridTree | SOLO  | FedTree     | SecureBoost | Pivot       | TFL   | ALL-IN |
|------------|-----------|-----------|-------|-------------|-------------|-------------|-------|--------|
| #instances | 20%       | **0.778** | 0.598 | 0.712-0.73  | 0.712-0.732 | 0.709-0.728 | 0.723 | 0.794  |
|            | 50%       | **0.786** | 0.61  | 0.726-0.742 | 0.726-0.742 | 0.723-0.735 | 0.73  | 0.805  |
|            | 80%       | **0.814** | 0.636 | 0.749-0.776 | 0.745-0.776 | 0.723-0.754 | 0.762 | 0.832  |
|            | 100%      | **0.832** | 0.653 | 0.764-0.788 | 0.764-0.788 | 0.752-0.789 | 0.773 | 0.853  |
| #features  | 20%       | **0.602** | 0.424 | 0.541-0.561 | 0.541-0.561 | 0.535-0.558 | 0.542 | 0.621  |
|            | 50%       | **0.724** | 0.552 | 0.655-0.679 | 0.655-0.678 | 0.659-0.678 | 0.669 | 0.742  |
|            | 80%       | **0.771** | 0.592 | 0.704-0.735 | 0.704-0.735 | 0.698-0.728 | 0.713 | 0.791  |
|            | 100%      | **0.832** | 0.653 | 0.764-0.788 | 0.764-0.788 | 0.762-0.784 | 0.773 | 0.853  |

## C.10 RESULTS OF COD-RNA

The results of cod-rna with different numbers of guests and levels of heterogeneity are presented in Figure 9, corresponding to Section 5.4 and Section C.3 of the main paper. HybridTree still outperforms the other baselines on this dataset.

## D DISCUSSIONS

**Broader Impact** Federated learning offers a compelling avenue that fosters multi-party collaboration, and our approach propels this direction a step further. Our approach encourages collaborative learning between heterogeneous parties to provide better services for people while preserving data privacy. However, our approach rests upon the premise of trust amongst the participating parties. In circumstances where collusion among multiple parties occurs, there lies the potential risk of inferring sensitive information from other parties. Thus, ensuring the integrity of the system is paramount prior to any real-world deployment.

**Limitations** Our method is well-suited for tabular data, as it allows for the representation of knowledge through meta-rules. However, when dealing with image, text, and graph data, the knowledge inherent in these types of data often cannot be easily captured by rule-based expressions, rendering our method less applicable in those cases. One interesting future direction is to combine deep neural networks (DNNs) with trees, where DNNs are trained locally to extract the low-dimensional representations, and trees are trained in the federated setting to classify the representations. This hybrid approach holds promise for addressing the challenges associated with image, text, and graph data in the context of hybrid federated learning.

**Related Work** We have summarized the related work on federated GBDTs in Section 2.2. Here we compare HybridTree and related federated GBDT studies in Table 13. We can observe that HybridTree is the first federated GBDT algorithm on hybrid data setting. Moreover, instead of using node-level or tree-level knowledge aggregation in existing studies, HybridTree adopts layer-level aggregation that effectively and efficiently incorporates the knowledge of guest parties by appending layers, which makes it more practical.

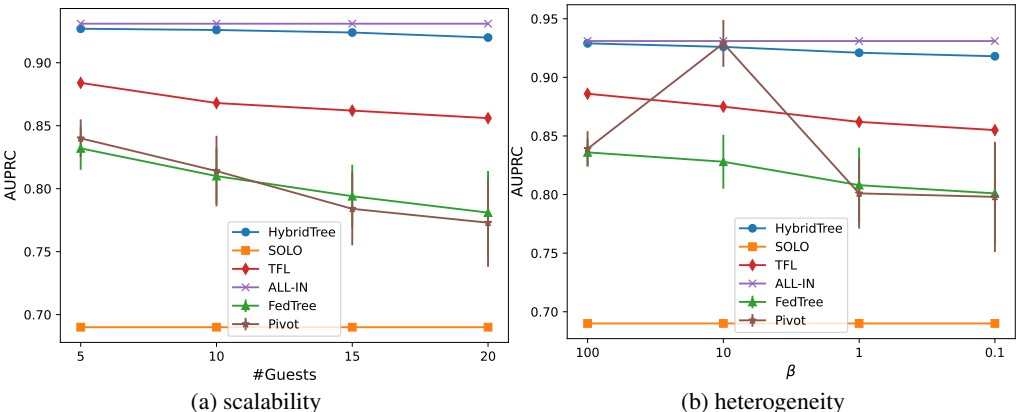

Figure 9: Experiments on scalability and heterogeneity of HybridTree on Cod-rna. We omit Secure-Boost as its curve overlaps with FedTree.

Table 13: Comparison between HybridTree and other federated GBDT studies.

| | Setting | | | Knowledge aggregation | | |
|---|---|---|---|---|---|---|
| | Horizontal | Vertical | Hybrid | node-level | tree-level | layer-level |
| SecureBoost (Cheng et al., 2019) | ✗ | ✓ | ✗ | ✓ | ✗ | ✗ |
| FedTree (Li et al., 2023) | ✓ | ✓ | ✗ | ✓ | ✗ | ✗ |
| Federboost (Tian et al., 2020) | ✓ | ✓ | ✗ | ✓ | ✗ | ✗ |
| TFL (Zhao et al., 2018) | ✓ | ✗ | ✗ | ✗ | ✓ | ✗ |
| SimFL (Li et al., 2020) | ✓ | ✗ | ✗ | ✗ | ✓ | ✗ |
| Secure XGB (Fang et al., 2021) | ✗ | ✓ | ✗ | ✓ | ✗ | ✗ |
| Pivot (Wu et al., 2020) | ✗ | ✓ | ✗ | ✓ | ✗ | ✗ |
| Feverless (Wang et al., 2022) | ✗ | ✓ | ✗ | ✓ | ✗ | ✗ |
| FBDT-DP (Maddock et al., 2022) | ✓ | ✗ | ✗ | ✓ | ✗ | ✗ |
| HybridTree | ✗ | ✓ | ✓ | ✗ | ✗ | ✓ |

