# OpenReview forum: "Effective and Efficient Federated Tree Learning on Hybrid Data"
_ICLR.cc/2024/Conference — ICLR 2024 poster_

### Official Review · Reviewer_2kKa · 2023-10-28

**Soundness:** 3 good
**Presentation:** 2 fair
**Contribution:** 4 excellent
**Rating:** 8
**Confidence:** 3

**Summary:**

The paper addresses federated learning with hybrid tabular data, where data from different parties may differ both in the features
and samples. The common framework for this kind of settings is federated learning using gradient boosting decision trees (GBDT). However, existing solutions suffer from high communication and computation overhead. The paper introduces HybridTree, a GBDT-based approach with some clever design. The higher layers of a HybridTree follows a standard GBDT design, exploring host features. The lower layers of the HybridTree are distributed among the guests. This is a form of bagging where each guest contributes one tree to the bag, utilising only the local data of the guest. At each HybridTree training iteration, there are $k$ bags corresponding to $k$ endpoints of the host (sub)tree.
Experiments on simulated and natural hybrid federated datasets show that the HybridTree can be up to 8 times faster than existing approaches, while maintaining the level of accuracy close to when all data are not protected by privacy and low communication overhead.

**Strengths:**

The paper has good motivation and good overview. The first part of the paper consists of a demonstration of the existence and abundance of meta rules in federated learning using GBDT, which provides clues about where to improve upon existing approaches, and the theoretical contribution, i.e. theorem 2 and 3, albeit being rather simple and straightforward from the mathematical point of view, provides an interesting view on how the following HybridTree algorithm tackles the inefficiency of existing approaches.

The second part of the paper, the introduction of the HybridTree algorithm, in my view is a clever and efficient use of boosting and bagging under the context of privacy preserving of federated learning. HybridTree is like 2D boosting with 1 dimension resembling GBDT addressing global (host) data, and with the other dimension spanning across local (guest) data, eliminating guest-guest communications and mitigating guest-host communications.

Experiments on both simulated and hybrid federated datasets are appear adequate for the problem that the paper addresses, with results favouring HybridTree with reasonable gaps over existing approaches.

**Weaknesses:**

I only have a few minor complaints about presentation, which makes it difficult to follow in the first read:

1. Definition 1 needs revising. At the beginning $S$ is defined as an intersection of split conditions but at the end you have split conditions not in $S$. It is intuitively understandable but it is mathematically incorrect.

2. In Figure 2b, you could have mentioned that $(L'_1, L'_2, L'_3, L'_4)$ can be $(L_1, L_2, L_1, L_3)$ as an example, and the audience would understand Theorem 2 better.

3. I did not understand theorem 3 at first. How does a meta rule that "ended by $F_g$ differ from a meta rule where $F_g$ is the last layer? Aren't they the same thing? Does the intersection operator here suggest ordering as well? After reading the proof of theorem 3, I understood what the author(s) meant. In any path from the root node to a leaf node, there is a possibility that a guest split condition stays above a host split condition, you just want transform the tree so that such possibility does not exist, is that right? If so, I suggest to restate theorem 3 in a way that is more understandable.

4. What has caught me a surprise is that theorem 3 is not used in a conventional way. Instead of using theorem 3 to improve upon a given tree, the authors redesign a new tree where guest split conditions are always below host split conditions. This is a good point that appears to have been presented somewhat lightly in the paper.

5. In my view, the flow of the paper could better if the authors presented the HybridTree inference before presenting the HybridTree training. Once the audience knows how inference works, the training components make much more sense.

**Questions:**

I do not have any major question.

---

> ### Author Response · Authors · 2023-11-18
> **Thanks for your comments!**
>
> > Q1. Definition 1 needs revising. At the beginning $S$ is defined as an intersection of split conditions but at the end you have split conditions not in $S$. It is intuitively understandable but it is mathematically incorrect.
>
> Thank you for pointing out this inconsistency in Definition 1. We have revised the definition to address this issue and ensure mathematical correctness. The revised definition is as follows:
>
> **Definition 1 (Meta-Rule)** Given a split rule $S:= \cap_{j=1}^N F_j$ where $F_j$ is a split condition, we call $S$ as a meta-rule if $P(y|x\in S) = P(y|x\in (S\cap F_k))$, $\forall F_k \neq F_j (j\in[1,N])$.
>
> > Q2. In Figure 2b, you could have mentioned that $(L'_1, L'_2, L'_3, L'_4)$ can be $(L_1, L_2, L_1, L_3)$ as an example, and the audience would understand Theorem 2 better.
>
> Thank you for the suggestion! We have revised the figure accordingly (now **Figure 3b**) in the revision.
>
> > Q3. In any path from the root node to a leaf node, there is a possibility that a guest split condition stays above a host split condition, you just want transform the tree so that such possibility does not exist, is that right? If so, I suggest to restate theorem 3 in a way that is more understandable.
>
> Yes, we want to transform the tree such that the guest split condition stays in the bottom layers. To clarify this transformation process, we have revised Theorem 3 as follows:
>
> **Theorem 3.** Suppose $S_{m}:=F_h \cap ... \cap F_g$ is a meta-rule in tree $\theta_A$ where $F_g$ is a split condition using the feature from the guests. For any tree path in tree $\theta_A$ involving the split nodes in $S_m$, we can always reorder the split nodes in the tree path such that $F_g$ is in the last layer. Moreover, naming the tree after the reordering as $\theta_B$, we have $E[f(\textbf{x}; \theta_A)] = E[f(\textbf{x}; \theta_B)]$ for any input instance $\textbf{x} \in \mathcal{D}$.
>
>
> > Q4. What has caught me a surprise is that theorem 3 is not used in a conventional way. Instead of using theorem 3 to improve upon a given tree, the authors redesign a new tree where guest split conditions are always below host split conditions. This is a good point that appears to have been presented somewhat lightly in the paper.
>
> We appreciate your observation regarding our Theorem 3. Indeed, we employ Theorem 3 as a foundational principle to design a new tree structure where guest split conditions are strategically positioned below those of the host. We have highlighted this point in the revision, including **Section 1, Section 3.2, and the beginning of Section 4**.
>
> > Q5. In my view, the flow of the paper could better if the authors presented the HybridTree inference before presenting the HybridTree training. Once the audience knows how inference works, the training components make much more sense.
>
> We think it may not be easy for some audiences to appreciate the inference process before introducing how the tree is divided among the parties. To address your concern, we have added a paragraph to briefly introduce the training and inference processes at the beginning of **Section 4** in the revision. In this paragraph, we mentioned that each tree is divided into multiple parties, and the host and guest parties collaboratively build the split path of an input instance and make the prediction. Given this background, the audience can understand the training component more easily.

---

> > ### Comment · Reviewer_2kKa · 2023-11-19
> > **Thank you!**
> >
> > Thank you for your feedback. I'd like to keep my rating.

---

> > > ### Author Response · Authors · 2023-11-19
> > >
> > > Thank you so much for supporting our work! Your constructive suggestions have greatly improved our paper.

---

### Official Review · Reviewer_whfr · 2023-10-31

**Soundness:** 3 good
**Presentation:** 2 fair
**Contribution:** 3 good
**Rating:** 5
**Confidence:** 2

**Summary:**

This paper introduces HybridTree, a novel federated learning approach tailored for hybrid data environments, where data from different parties vary in both features and samples.  HybridTree facilitates federated tree learning by capitalizing on consistent split rules observed in trees, allowing the integration of knowledge from different parties into the lower layers of a tree. Using these insights, the paper proposes a layer-level solution to train trees without the need for heavy communication. Experimentally, HybridTree has shown comparable accuracy to centralized models while substantially reducing computational and communication overhead. In comparison to other baselines, it can achieve up to an 8-fold speedup.

**Strengths:**

1. The problem studied in this paper is interesting and important. This work proposes federated tree models on hybrid data, which expands the scope of current FL frameworks and makes an important contribution to the FL community.
2. This paper is original and technically sound. The main claims regarding the proposed setting are well supported in the methodology and experimental parts.

**Weaknesses:**

1. The paper is not easy to follow.
2. It is better to provide more analysis or explanation on the training process in section 4.1 to let readers well understand how HybridTree handles hybrid data and makes their contribution to the improvement.
3. Although there is a relatively thorough literature review in the related work part, I prefer to see a discussion on the relation of this work, especially the specific methods.
4. Some minor errors, see below.

**Questions:**

1. "Eq 2" —> "Eq (2)"
2. $D_{G_i}= \{ x \} (x \in R^{ d_{g_i}})$ —> $D_{G_i}= \{ x | x \in R^{ d_{g_i}}\} $,  $D_H= { x, y } (x \in {R^{d_h}})$ —> $D_H= \{ x, y | x \in R^{d_h}\} $
3. "Last, guests update the following lower layers of the tree using their local features and received encrypted gradients and send back the encrypted prediction values." —> "Last, guests update the following lower layers of the tree using their local features and **receive** encrypted gradients**,** and send back the encrypted prediction values."
4. "Then, the gain of the split value is defined by the loss reduction after split, which is" —>"Then, the gain of the split value is defined by the loss reduction after **the split**, which is"
5. "The best-split point is selected among these split candidates. When reaching the maximum depth or the gain is always negative, the current node will become a leaf node." —> "The **best split** point is selected from these candidates. If the tree reaches the maximum depth or **if** the gain remains negative, the current node becomes a leaf node."
6. "FL on hybrid data is rarely exploited in the current literature. Zhang et al. (2020) proposes to…. Liu et al. (2020) applies transfer" —> "Zhang et al. (2020) **propose** to…. Liu et al. (2020) **apply** transfer"

7. “For simplicity, we start from a single-host with multi-guest setting” —>  “For simplicity, we start from a single-host with multi-guest setting” —> "For simplicity, we start from a scenario involving a single host with multiple guests."

---

> ### Author Response · Authors · 2023-11-18
> **Thanks for your comments!**
>
> > Q1. The paper is not easy to follow.
>
> We apologize for any difficulties encountered in following the initial version of our paper and appreciate your feedback on this aspect. We have improved the writing in the revision, including 1) We have updated the problem statement and added **Figure 1** to clarify the hybrid data setting more clearly. 2) We have improved **Definition 1, Theorem 3, and Figure 3(b)** to make our tree transformation theorem easier to understand. 3) We have added a paragraph to introduce the training and inference process of our algorithm in general at the beginning of **Section 4** to improve readability. 4) We have added a paragraph and added more annotations in Algorithm 1 to further demonstrate the training process of our algorithm in **Section 4.1**. 5) We have added a table to summarize the notations used in the paper in **Table 4 of Appendix B.1**.
>
> > Q2. It is better to provide more analysis or explanation on the training process in section 4.1 to let readers well understand how HybridTree handles hybrid data and makes their contribution to the improvement.
>
> We appreciate your suggestion for a more detailed explanation of the training process, especially in the context of handling hybrid data. In response, we have enriched **Section 4.1** with an additional paragraph that delves into the nuances of the training process of HybridTree. Furthermore, we have enhanced Algorithm 1 with more detailed annotations for greater clarity.
>
> In general, there are three steps in the whole training process: 1) The host party updates a subtree individually (Lines 1-9); 2) The host party sends the encrypted intermediate results to the guest parties (Lines 10-13); 3) The guest parties update the bottom layers individually and send back the encrypted prediction values (Lines 14-21). Since HybridTree does not require accessing all features and instances when updating each node, it can handle the hybrid data case where each party only has partial instances and features. Moreover, based on our analysis in Section 3, by updating the bottom layers using the guests' features, the meta-rule knowledge of the guest parties can be effectively incorporated.
>
> > Q3. Although there is a relatively thorough literature review in the related work part, I prefer to see a discussion on the relation of this work, especially the specific methods.
>
> Thank you for your suggestion to delve deeper into the comparative aspects of HybridTree in relation to existing federated GBDT studies. We have added the comparison between HybridTree and related federated GBDT studies in **Table 13 of Appendix D**.
>
> 1) Novel Setting for Federated Learning: HybridTree introduces an innovative approach tailored for the hybrid data setting. This setting, which has not been extensively explored in previous federated GBDT studies, is increasingly relevant in diverse real-world applications where data characteristics vary significantly across different parties.
>
> 2) Layer-Level Knowledge Aggregation: Contrasting the common node-level or tree-level knowledge aggregation methods found in existing studies, HybridTree employs a layer-level aggregation approach. This method allows for a more efficient and effective incorporation of knowledge from guest parties. By appending layers specific to each party’s data, HybridTree ensures that the model benefits from the diverse insights and characteristics inherent in the hybrid data.
>
> > Q4. Some minor errors, see below.
>
> We greatly appreciate your attention to detail in identifying these minor errors. We have reviewed the entire manuscript and corrected all the typos and inaccuracies you pointed out. To ensure thoroughness, we conducted an additional round of proofreading to identify and rectify any remaining errors.

---

> ### Author Response · Authors · 2023-11-21
> **Looking forward to your feedback**
>
> Dear Reviewer whfr,
>
> We have addressed all your comments in our response and revision. We have improved the writing of our paper according to your suggestions. The Rebuttal/Discussion phase will end soon. We appreciate it if you could read our rebuttal and provide any feedback. Thanks!
>
> Regards,
>
> Paper 768 authors

---

> > ### Author Response · Authors · 2023-11-22
> >
> > Dear Reviewer whfr,
> >
> > The Rebuttal/Discussion phase will end soon. Could you please read our rebuttal and provide any feedback? We have made a lot of effort in the revision. Thanks a lot!
> >
> > Regards,
> >
> > Paper 768 authors

---

> ### Comment · Reviewer_whfr · 2023-11-23
>
> Thanks for the authors' reply and revisions. I'd like to maintain my score.

---

### Official Review · Reviewer_3ksq · 2023-11-01

**Soundness:** 2 fair
**Presentation:** 2 fair
**Contribution:** 2 fair
**Rating:** 5
**Confidence:** 5

**Summary:**

This work proposed a novel tree learning protocol for the hybrid FL settings where data are distributed both by features and by samples, a scenario that is less explored. The proposed method does not need frequent communication between parties and exhibits comparable accuracy to the centralized setting.

**Strengths:**

1. This work targets at a hybrid FL setting, where vertical FL setting integrates with horizontal setting. The setting has practical applications in many industrial areas but are less studied in research.

2. The observation of the existence of meta-rule is novel and leads to the development of a communication efficiency tree-based FL algorithm.

**Weaknesses:**

1. The paper is not very well-presented and is hard to follow. First of all,  it is unclear in the hybrid setting considered, what are the relative relations of the guest parties? In the introduction, it appears that they share the same feature space but have different sample IDs, however, in 3.1 they appear to have different dimensions and unclear alignment. It is suggested that the paper properly define the problem setting. A figure on how data is partitioned by different parties would also help.  Secondly, the algorithm is very messy with many undefined variables and notations, making it also hard to process. It is also not clear how features from different guests are used in a collaborative manner in the approach due to the above issues on presentation.

2. The experimental results are biased. 1) The paper compares its model performance using data from multiple guests against FedTree and Pivot using data from one guest, therefore the model performance gap is largely due to the utilization of additional data parties, not to the benefits of the algorithm. To be fair, the paper should compare them under the same settings, that is, multiple guests and host. 2) Important baselines such as Secureboost[1] is missing.  3) the model performance of the proposed methods still appear to be
a little inferior to the centralized setting, not exactly "comparable" as claimed. It is important to understand whether the proposed method is "lossless" or "lossy" and why. I think more detailed examinations and explanations are needed here.

[1] Cheng, Tao Fan, Yilun Jin, Yang Liu, Tianjian Chen, Dimitrios Papadopoulos, and Qiang Yang. Secureboost: A lossless federated learning framework, 2019

**Questions:**

1. Does a "meta-rule" have to include the last layer of a tree? Fig 1& 2 both show that they include the last layer. However, Definition 1 shows it has additional layers F_k, which seems to be inconsistent.

2. How does the algorithm deal with trees that have mixed splits from both host party and guest party? Figure 3 only demonstrates a simple example that the guest splits appear in the bottom layers, but what if guest splits appear near the root, followed by intervened and alternative host and guest splits? More complicated examples will help to demonstrate how the framework works.

---

> ### Author Response · Authors · 2023-11-18
> **Thanks for your comments!**
>
> > Q1. The paper is not very well-presented and is hard to follow. First of all, it is unclear in the hybrid setting considered, what are the relative relations of the guest parties? In the introduction, it appears that they share the same feature space but have different sample IDs, however, in 3.1 they appear to have different dimensions and unclear alignment. It is suggested that the paper properly define the problem setting. A figure on how data is partitioned by different parties would also help.*
>
> We apologize for any confusion caused by our presentation. Our primary focus is on scenarios where guest parties share the same feature space but possess different sample IDs. Examples include a payment system across multiple banks or a hospital network utilizing patient data from various sources such as iWatches. We have revised **Section 3.1**, concentrating on the homogeneous guest feature space setting to prevent ambiguity. Additionally, we have added **Figure 1** in the revised version, illustrating the data partitioning among different parties. This visualization should provide a clearer understanding of our setting.
>
> Though our focus is on setting where guest parties share the same feature space but different sample IDs, our algorithm does not pose a limitation on the feature and sample space of the guest parties. In **Appendix C.3**, we also provide experimental results where the guest parties have different feature spaces and overlapped samples, where our approach still outperforms the other baselines.
>
> > Q2. The algorithm is very messy with many undefined variables and notations, making it also hard to process. It is also not clear how features from different guests are used in a collaborative manner in the approach due to the above issues on presentation.
>
> Thank you for pointing out it. To enhance clarity, we have added **Table 4 in Appendix B**, which summarizes all the notations used in our algorithm. We have also expanded the annotations in Algorithm 1 for better understanding.
>
> Regarding the collaborative use of features from different guests, this is articulated in the GuestTrain function of Algorithm 1 (line 14, lines 16-21). Here, following the host party's update of a tree with its features and transmission of encrypted gradients and instance ID sets to the guest parties, these guests further expand the tree with deeper layers using their local features.
>
> We have added an additional paragraph to summarize the training process. In general, there are three steps in the whole training process: 1) The host party updates a subtree individually (Lines 1-9); 2) The host party sends the encrypted intermediate results to the guest parties (Lines 10-13); 3) The guest parties update the bottom layers individually and send back the encrypted prediction values (Lines 14-21). Since HybridTree does not require accessing all features and instances when updating each node, it can handle the hybrid data case where each party only has partial instances and features. Moreover, based on our analysis in Section 3, by updating the bottom layers using the guests' features, the meta-rule knowledge of the guest parties can be effectively incorporated.
>
> > Q3. The paper compares its model performance using data from multiple guests against FedTree and Pivot using data from one guest, therefore the model performance gap is largely due to the utilization of additional data parties, not to the benefits of the algorithm. To be fair, the paper should compare them under the same settings, that is, multiple guests and host.
>
> As claimed in the paper, to the best of our knowledge, we are **the first ones to develop the federated GBDT algorithm on hybrid data**. While FedTree and Pivot support the vertical FL setting, they cannot handle the hybrid data setting with multiple guests. Another baseline TFL utilizes the data from all parties and adopts a tree-level aggregation. HybridTree significantly outperforms TFL as shown in our experiments.

---

> > ### Comment · Reviewer_3ksq · 2023-11-22
> >
> > I have read the authors' responses, and appreciate their efforts to improve the presentation of the paper. However, the rebutal didn't address my concerns on the biased comparisons on model performance with existing works. The current results are still very misleading to demonstrate its superior model performance over other tree algorithms such as FedTree and Secureboost. The authors compare its performance using data from multiple guest parties (hybrid setting) to these works using only one guest party (non-hybrid setting). So the performance gain is due to the larger data size of the hybrid setting, not the algorithm itself.  To be fair, the authors should compare all these algorithms under the same number of parties, which includes 1) 2-party VFL (1 guest and 1 host), and 2) multi-party VFL(N guests and 1 host). It is not true that other algorithms such as Secureboost can not handle hybrid data, as they can be readily applied to these settings by aggregating the guests' statistical knowledge, just as the paper did for its aggregation step. In fact, the paper's main contribution is, as it stated itself, "it does not need frequent communication traffic to train a tree." In order to accomplish that, the algorithm is "lossy", i.e., model performance is compromised, as shown by comparing its results to the model performance of centralized training, whereas other algorithms such as Secureboost do not sacrifice model performance. I think an experiment under a 2-party VFL setting would demonstrate that.  I would suggest the paper focus on its evaluations of efficiency-utility tradeoff over a broad range of number of guest parties  (starting from 1), rather than claiming itself as the first algorithm that can handle hybrid data.  Due to these concerns, I will keep my score.

---

> ### Author Response · Authors · 2023-11-18
> **Thanks for your comments!**
>
> > Q4. Important baselines such as Secureboost [1] are missing.
>
> Initially, we excluded SecureBoost based on its similarity in model performance with FedTree, as both employ a comparable node-level aggregation design, which often results in overlapping model performance [2]. We have now included SecureBoost in our revised version.
>
> SecureBoost has been added to the experimental comparisons in **Tables 1-4, Table 6, and Tables 8-12**. To maintain clarity and readability in our graphical representations, we opted not to include SecureBoost in figures such as Figure 6, where its performance curve closely aligns with that of FedTree. Our updated results confirm that FedTree's performance is indeed very similar to SecureBoost. Moreover, the efficiency of HybridTree also significantly outperforms SecureBoost.
>
> [2] FedTree: A Federated Learning System For Trees
>
> > Q5. The model performance of the proposed methods still appear to be a little inferior to the centralized setting, not exactly "comparable" as claimed. It is important to understand whether the proposed method is "lossless" or "lossy" and why. I think more detailed examinations and explanations are needed here.
>
> We appreciate your observation regarding the performance comparison with the centralized setting. Our method, indeed, does not achieve a completely lossless transfer compared to centralized solutions, and there are two primary reasons for this: 1) While our tree transformation in Section 3 ensures that the expected prediction value for any input remains unchanged, this mathematical expectation is an average measure. In practice, the prediction value for each individual sample may not align perfectly with this expected value, leading to some discrepancies. 2) Our algorithm is designed to integrate meta-rule knowledge through the training of additional layers by guest parties. As illustrated in Figure 3(a), this meta-rule knowledge is significant. However, certain aspects of knowledge contributed by guest parties are not encapsulated by meta-rules (e.g., about 10% of trees do not have meta-rules in AD). While our algorithm efficiently aggregates meta-rule knowledge, it does so at the cost of some accuracy to enhance overall practicality and efficiency.
>
> > Q6. Does a "meta-rule" have to include the last layer of a tree? Fig 1& 2 both show that they include the last layer. However, Definition 1 shows it has additional layers F_k, which seems to be inconsistent.
>
> Yes, the meta-rule has to include the last layer of a tree. To enhance clarity, we have revised **Definition 1** in the manuscript. We define a split rule as a meta-rule as long as the prediction is determined if its features satisfy the split rule, no matter what additional feature constraints are given. In Definition 1, $F_k$ is not an additional layer. Instead, it refers to any potential split condition that is not part of the meta-rule. We use $P(y|x\in S) = P(y|x\in (S\cap F_k))$ to formulate the deterministic of the prediction given the meta-rule.
>
> **Definition 1 (Meta-Rule)** Given a split rule $S:= \cap_{j=1}^N F_j$ where $F_j$ is a split condition, we call $S$ as a meta-rule if $P(y|x\in S) = P(y|x\in (S\cap F_k))$, $\forall F_k \neq F_j (j\in[1,N])$.
>
> > Q7. How does the algorithm deal with trees that have mixed splits from both host party and guest party? Figure 3 (now Figure 4 in the revision) only demonstrates a simple example that the guest splits appear in the bottom layers, but what if guest splits appear near the root, followed by intervened and alternative host and guest splits? More complicated examples will help to demonstrate how the framework works.
>
> Figure 3 has been relabeled as Figure 4 in our revision. Note that the design of our algorithm is to only use the guest features in the bottom layers as presented in Figure 4b. This design is a deliberate strategy to minimize computation and communication overhead, a common challenge in existing federated GBDT algorithms that employ node-level aggregation across all parties (as shown in Figure 4a). Based on our analysis in Section 3, even though the guest splits appear near the root (e.g., Figure 3b tree A), we can still reorder it to the last layer while keeping the tree performance (e.g., Figure 3b tree B). In other words, training a tree where the guest features are in the last layers is sufficient to capture the meta-knowledge of the guests.
>
> We have added another more complicated tree transformation example in **Figure 7 in Appendix A**. In the figure, there are split paths with mixed splits from host and guest features in Tree A, and we can transform it into Tree B where the guest features are in the bottom layers.

---

> ### Author Response · Authors · 2023-11-21
> **Looking forward to your feedback**
>
> Dear Reviewer 3ksq,
>
> We have addressed all your comments in our response and revision including the presentation and the experiments. We noticed that you may have some misunderstandings about our proposed algorithm. The Rebuttal/Discussion phase will end soon. We appreciate it if you could read our rebuttal and provide any feedback. Thanks!
>
> Regards,
>
> Paper 768 authors

---

> ### Author Response · Authors · 2023-11-22
>
> Thanks for your feedback.
>
> > Q1. It is not true that other algorithms such as Secureboost can not handle hybrid data, as they can be readily applied to these settings by aggregating the guests' statistical knowledge, just as the paper did for its aggregation step.
>
> **Our paper does not aggregate the guest statistical knowledge in each node. We cannot apply what we did in HybridTree to SecureBoost.** As presented in Section 4.1, in our algorithm, thanks to our layer-wise design, each guest individually trains several layers, and only the prediction values of the guest parties are averaged (Line 15). For SecureBoost, it cannot be directly extended to the hybrid data setting as it needs the aggregation of the gradients in each node. Algorithm 1 of [1] requires aggregating the encrypted gradient statistics. In the multi-guest setting, since each guest has different samples, how to propose the split points so that each guest party has the same split points while not leaking data privacy? If the guest parties do not have the same split points, how to aggregate their gradients? For the TFL baseline, since it adopts a tree-level aggregation, we extend it to the hybrid data setting in our evaluation and the results show that HybridTree achieves a higher model performance than TFL.
>
> [1] SecureBoost: A Lossless Federated Learning Framework. https://arxiv.org/pdf/1901.08755.pdf
>
> > Q2. I think an experiment under a 2-party VFL setting would demonstrate that.
>
> **We indeed presented the experiments in Table 10 and Table 11 of the paper.** HybridTree achieves lower model performance than VFL studies, and the performance gap is about 0.01. In the meantime, the speedup of HybridTree is significant, which is over 30x against SecureBoost.
>
> > Q3. I would suggest the paper focus on its evaluations of efficiency-utility tradeoff over a broad range of number of guest parties (starting from 1), rather than claiming itself as the first algorithm that can handle hybrid data.
>
> As claimed in the response to Q1, we cannot find a non-trivial way to extend the VFL studies to the hybrid data setting. Our experiments show that HybridTree is much more efficient than VFL studies, even though they utilize less data than HybridTree.

---

> > ### Author Response · Authors · 2023-11-23
> >
> > FYI, we have made the following minor revisions to the paper:
> > * We have highlighted that the VFL baselines are on the 2-party setting and it is non-trivial to extend them to the hybrid data setting in the second paragraph of Section 5.1.
> > * We have highlighted that the TFL baseline is conducted on the hybrid data setting with all parties in the second paragraph of Section 5.1.
> > * We have changed the claim of "the first algorithm that can handle hybrid data" to "a new algorithm that can handle hybrid data".
> >
> > Also, if you have any concrete ideas about how to easily extend SecureBoost to our hybrid data setting, please let us know. Thanks!

---

> > > ### Comment · Reviewer_3ksq · 2023-11-23
> > >
> > > I appreciate the authors' clarifications and revisions to the paper. The results on 2-party setting are helpful, as it demonstrates that the algorithm is indeed "lossy" compared to other tree algorithms. I think the paper should be very clear about it instead of claiming that its performance is "comparable" to centralized training. As for extending Secureboost to hybrid data setting, since in Secureboost  at every node the active party enumerates all the proposed splits and finds the optimal one at the active party, meaning that it is reasonable to assume that guests and active parties can establish consensus on proposed splits (e.g. based on quantiles or prior knowledge on feature distributions). Therefore I don't find it a particular obstacle for not being able to establish proposed splits and extend given this assumption. Comparing it will further help understand the communication savings in practical cases.
> > > Nevertheless, given the current revised manuscript, I decide to raise my scores.

---

### Official Review · Reviewer_KFHq · 2023-11-06

**Soundness:** 3 good
**Presentation:** 3 good
**Contribution:** 3 good
**Rating:** 6
**Confidence:** 3

**Summary:**

The paper presents HybridTree, an innovative algorithm tailored for federated learning within the framework of Gradient Boosting Decision Trees (GBDT) in a hybrid data context. It introduces a novel tree transformation technique that can reorder split features based on insights from meta-rules. This transformation is key to the development of the hybrid tree learning algorithm, which is capable of incorporating knowledge from different sources (guests) by appending new layers to the decision trees.

**Strengths:**

1. The main innovation of this work lies in the development of a tree transformation strategy that can reorder split features to accommodate a federated learning environment. This is particularly relevant for scenarios where data privacy and distribution are concerns.

2. By introducing a new layer-level training algorithm, HybridTree, they address the integration of knowledge from multiple participants (referred to as "guests") in the federated model without compromising on data privacy.

3. The authors conduct extensive experiments on multiple datasets. The empirical results show that the proposed method outperforms the baselines significantly in model performance.

**Weaknesses:**

Even though the proposed method can handle hybrid features, the features have to be tabular data. This might not be the constraint of this paper but rather the limitation of the tree-based methods. However, maybe the authors can consider the scenarios where clients have multi-modal data, where the data modalities are hybrid across clients.

**Questions:**

Please see the weakness.

---

> ### Author Response · Authors · 2023-11-18
> **Thanks for your comments!**
>
> > Q1. Even though the proposed method can handle hybrid features, the features have to be tabular data. This might not be the constraint of this paper but rather the limitation of the tree-based methods. However, maybe the authors can consider the scenarios where clients have multi-modal data, where the data modalities are hybrid across clients.*
>
> Thank you for highlighting this significant aspect. Indeed, our current method focuses on tabular data, primarily due to the inherent structural limitations of tree-based methods when dealing with multi-modal data. The scenario you've mentioned, involving clients with hybrid multi-modal data, presents a very interesting and challenging direction. We consider it as a future work using models besides trees.

---

> > ### Comment · Reviewer_KFHq · 2023-11-21
> > **Thanks for the reply**
> >
> > Thanks for the authors' reply. I'd like to keep my score.

---

> > > ### Author Response · Authors · 2023-11-21
> > >
> > > Thank you so much for supporting our work!

---

### Author Response · Authors · 2023-11-18
**Summary of Revision**

Dear Reviewers,

We sincerely appreciate the time and effort you have dedicated to reviewing our manuscript. Your constructive comments have been instrumental in enhancing the quality of our work. In response, we have revised our paper to address each of your concerns. The key changes in our revision are outlined below and are also highlighted in $\textcolor{red}{red}$ in the revised manuscript for ease of reference:
* We have updated our main contributions to highlight the importance of our tree transformation method in **Section 1** (Reviewer $\textcolor{red}{2kKa}$).
* We have updated the problem statement and added **Figure 1** to clarify the hybrid data setting more clearly (Reviewer $\textcolor{blue}{3ksq}$, $\textcolor{orange}{whfr}$).
* We have improved **Definition 1, Theorem 3, and Figure 3(b)** to make our tree transformation theorem easier to understand (Reviewer $\textcolor{blue}{3ksq}$, $\textcolor{orange}{whfr}$, $\textcolor{red}{2kKa}$).
* We have added a paragraph to introduce the training and inference process of our algorithm in general at the beginning of **Section 4** to improve readability (Reviewer $\textcolor{orange}{whfr}$).
* We have added a paragraph and added more annotations in Algorithm 1 to further demonstrate the training process of our algorithm in **Section 4.1** (Reviewer $\textcolor{blue}{3ksq}$, $\textcolor{orange}{whfr}$, $\textcolor{red}{2kKa}$).
* We have added an additional baseline SecureBoost in the experiments, including **Tables 1-3, Table 6, and Tables 8-12** (Reviewer $\textcolor{blue}{3ksq}$).
* We have added a future work in **Section 6** (Reviewer $\textcolor{purple}{KFHq}$).
* We have added an additional figure to demonstrate the tree transformation in **Figure 7 of Appendix A** (Reviewer $\textcolor{blue}{3ksq}$).
* We have added a table to summarize the notations used in the paper in **Table 4 of Appendix B.1** (Reviewer $\textcolor{blue}{3ksq}$, $\textcolor{orange}{whfr}$).
* We have added a comparison between our approach with related studies in **Table 13 of Appendix D** (Reviewer $\textcolor{orange}{whfr}$).

We look forward to your further feedback!

Regards,

Paper 768 authors

---

### Meta-Review · Area_Chair_wd1j · 2023-12-17

**Metareview:**

An interesting paper based on a simple observation (new def 1) which turns out to be useful in a federated learning setting. I am inclined to accept the paper based on several updates from the authors that have contributed to help reviewers and myself understand better the paper's point. I also subscribed to the value of tabular data per se and does not consider it to be in fact a limitation of the approach (KFHq).

This being said, I do consider that assuming that record linkage is known in advance is the real limitation of the work, because in the tabular data setting, small errors can have big impacts (change one feature and you can radically change a DT solution, and even if it does not, it can change the solution to record linkage in a way that radically impacts trees, see ICML'19 papers on this).

Nevertheless, the problem addressed is a real challenge and the paper proposes an original approach. Still, it is important that the authors further polish their paper to address key remarks (KFHq, SecureBoost for 3ksq, 2kKa).

**Justification For Why Not Higher Score:**

Limitation due to the fact that record linkage is known in advance (barely ever the case, but very challenging and a separate problem)

**Justification For Why Not Lower Score:**

The idea the paper is based on brings nice leverage for federated learning

---

### Decision · Program_Chairs · 2024-01-16

Accept (poster)